# RoboPoint: A Vision-Language Model for Spatial Affordance Prediction for Robotics

**Wentao Yuan**[1]    **Jiafei Duan**[1]    **Valts Blukis**[2]    **Wilbert Pumacay**[4]    **Ranjay Krishna**[1,3]

**Adithyavairavan Murali**[2]         **Arsalan Mousavian**[2]         **Dieter Fox**[1,2]

[1]University of Washington                    [2]NVIDIA
[3]Allen Institute for Artifical Intelligence          [4]Universidad Católica San Pablo

**Abstract:** From rearranging objects on a table to putting groceries into shelves, robots must plan precise action points to perform tasks accurately and reliably. In spite of the recent adoption of vision language models (VLMs) to control robot behavior, VLMs struggle to precisely articulate robot actions using language. We introduce an automatic synthetic data generation pipeline that instruction-tunes VLMs to robotic domains and needs. Using the pipeline, we train RoboPoint, a VLM that predicts image keypoint affordances given language instructions. Compared to alternative approaches, our method requires no real-world data collection or human demonstration, making it much more scalable to diverse environments and viewpoints. In addition, RoboPoint is a general model that enables several downstream applications such as robot navigation, manipulation, and augmented reality (AR) assistance. Our experiments demonstrate that RoboPoint outperforms state-of-the-art VLMs (GPT-4o) and visual prompting techniques (PIVOT) by 21.8% in the accuracy of predicting spatial affordance and by 30.5% in the success rate of downstream tasks. Project website: robo-point.github.io.

**Keywords:** Foundation Model, Affordance Prediction, Open-world Manipulation

## 1 Introduction

Spatial reasoning is fundamental to all intellectual processes [1]. Beyond its prominence in understanding geometry, science, and architecture [2], spatial reasoning significantly impacts our everyday lives. Even mundane tasks like purchasing groceries require us to identify the vacant space in our shopping carts to load more items. One critical mechanism through which we communicate plans that involve navigation and manipulation is by *pointing*. Studies in developmental psychology demonstrate that infants and adults alike point to share information about their environment [3]. In robotics, pointing has been operationalized through waypoints for navigation and task execution. Roboticists have found that when robots use waypoints effectively, it mimics human pointing, leading to more intuitive plans [4].

Recent explorations have cast aside pointing in favor of language instructions with the advent of large VLMs [5, 6, 7]. Trained on large datasets of images and texts, VLMs can provide visual semantic understanding and guidance to robotic tasks, such as which object a manipulator should pick up or which goal a mobile robot should reach [8, 9, 10]. However, language is not precise enough to successfully guide robot behavior. Even the most powerful VLMs, such as GPT-4o [11], have limited accuracy in real robot execution, especially when language commands contain spatial relations to identify objects or refer to object-free locations, such as "place the cup next to the plate".

In this work, we introduce RoboPoint, an open-source VLM instruction-tuned to *point*. Specifically, we fine-tune a pre-trained language model to perform *spatial affordance prediction*, the task of pointing at where to act according to language instructions. Two key features differentiate RoboPoint from other VLMs for robotics: a **point-based action space** and a **scalable data pipeline**. First, inspired by prior works [12, 13, 14], the actions are specified via points in the RGB image,

8th Conference on Robot Learning (CoRL 2024), Munich, Germany.

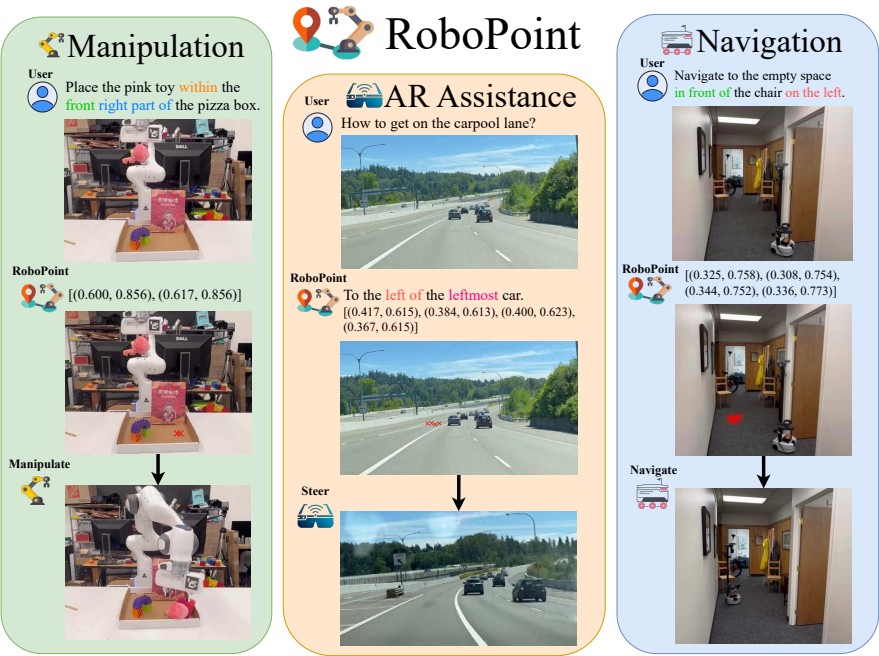

Figure 1: ROBOPOINT is a **Vision-Language Model** that predicts **affordance points** based on language instructions. It is able to generate precise actions (red crosses in the image) which satisfy spatial relations in the instruction. ROBOPOINT is a generic VLM that can be applied to many domains such as manipulation, augmented reality and navigation.

and then transformed to 3D using depth information, removing the need for pre-defined action primitives [9, 10], external object detectors [15, 16], or iterative visual prompting [17].

Second, we design a fully autonomous pipeline generating a large, diverse dataset of ground truth action points by computing spatial relations from the camera's perspective and sampling points within object masks and object-surface intersections. Compared to approaches that require expensive human demonstration data [18, 19, 20], our pipeline is much easier to scale. Even though we only added data containing simulated images along with templated language, the resulting model's performance improves on real images with natural language commands.

Our results show that ROBOPOINT significant outperforms various powerful VLMs such as GPT-4o [11], LLaVA-NeXT [21], Qwen-VL [6] and SpatialVLM [22] on relational object reference, free space reference and object rearrangement in cluttered, real-world environments, without losing accuracy on standard visual question answering (VQA) benchmarks. To evaluate relational free space reference, we collect WHERE2PLACE, a manually annotated, challenging real-world benchmark. We also show very promising results beyond robotic applications in an interactive augmented reality (AR) setting, where ROBOPOINT provides visual action suggestions, effectively guiding users through tasks by predicting target points based on common sense.

## 2 Related Work

Inspired by prior works on *spatial reasoning* and *affordance prediction*, ROBOPOINT takes a distinct approach to build VLMs for robotics in contrast to recent methods using *zero-shot language models*.

**Spatial Reasoning** Many VQA benchmarks [23, 24, 25, 26, 27, 28] have included problems about spatial relations as indicator for a model's ability to understand 3D. These problems can be solved using state estimation [29] plus symbolic reasoning [30], but these methods have poor generalization to novel objects. More recently, SORNet [31] shows that a transformer model conditioned on object prompts can generalize zero-shot to unseen objects on spatial reasoning tasks, similar in spirit to modern VLMs. However, existing works on spatial reasoning mostly focused on coarse-grained relations. SpatialVLM [22] took a step forward to predict spatial relations in metric space, but we show that ROBOPOINT can achieve better performance on real-world spatial reasoning tasks by locating affordances as points.

**Affordance Prediction** Affordance is defined as the functions of an object, i.e. in what ways it can be manipulated. It goes beyond the visual properties and ties observations to actions. The efficacy of affordance prediction has been shown by many learning-based manipulation methods for 6-DoF grasping [32, 33, 34] and stable object placement [35, 36, 37]. Affordance can be represented in many ways such as part segmentation [38], dense image feature descriptors [39] and keypoints [40, 41, 12]. We use the 2D keypoint representation to train ROBOPOINT since it can be readily converted into language format.

**Zero-shot Language Models for Robotics** Several works [8, 9, 10] have shown that language model are capable planners for robotic tasks. Using in-context learning [42], these methods generate reasonable plans in structured language, but require pre-defined action primitives to execute. More recent works leverage VLMs to generate more fine-grained outputs. VoxPoser [15] generates 3D value maps. PIVOT [17] iteratively samples and evaluates possible actions in image space. MOKA [16] predicts keypoints specific to an action type. Unlike ROBOPOINT, all of these approaches still rely on external models for detecting objects relevant for the task.

## 3 Method

ROBOPOINT is instruction-tuned from Vicuna-v1.5-13B [43] with a mix of synthetic and real-world data on spatial affordance prediction. This section will cover 3 critical aspects of the tuning pipeline: 1) the problem formulation 2) the instruction tuning procedure and 3) the curation of the data mix.

**Spatial Affordance Prediction** We formulate the problem of spatial affordance prediction as predicting a set of target point coordinates $\{(x_0, y_0), (x_1, y_1), ..., (x_n, y_n)\}$ in image space that satisfy the relations indicated by a language prompt. This formulation has several advantages. First, compared to fuzzy language actions such as "place the apple in the drawer", which require detection of apple and drawer before execution, a point prediction is much more precise and can be directly converted to actions. Most VLMs are trained to predict bounding boxes. However, from Fig. 3, we can see that bounding boxes often include a lot of undesirable clutter due to camera perspective and are not as specific as point outputs. Second, our formulation is general enough to enable various robotic tasks. For example, the predicted points can be interpreted as waypoints for navigation, contact points for grasping or region proposals for placement. This not only allows the model to perform multiple tasks but also means it can be trained with multi-task data.

**Instruction Fine-tuning** Min et al. [44] has shown that rather than learning new tasks, in-context learning [42] works by activating patterns from the training data. Thus, instead of mining patterns from the non-public training dataset, we opt to build our own dataset (see Sec. 4) and fine-tune the language model's parameters. Specifically, we follow the instruction tuning pipeline in Liu et al. [7]. As shown in Fig. 2, the model consists of an image encoder, a MLP projector, a language tokenizer and a transformer language model. The image encoder processes the image into a set of tokens which are then projected by a 2-layer MLP into the same space as the language tokens. The multimodal tokens are concatenated and passed through the language transformer. All modules are initialized with pre-trained weights. The projector and the transformer weights are allowed to update while the vision encoder and tokenizer weights are frozen. The model is autoregressive and the objective is to predict the response tokens and a special token delineating the boundary between instruction and response. Our results (Table 2, Fig. 5) show that our instruction-tuned model achieves much higher precision than baselines using in-context learning [17, 22].

**Co-finetuning with Synthetic Data** We find that providing the appropriate mix of data is crucial to the model's performance on downstream tasks. As observed by Brohan et al. [19], co-training with a mix of robotic data and internet data ensures the model does not forget the knowledge it has learned during pre-training. Our dataset for fine-tuning consists of 4 different sources, as illustrated in Table. 1. The VQA data is a mix of 665K conversations from [45] where the model is asked to answer questions in natural language based on the input image. This ensures the model can reason in language. The LVIS data is converted from [46], where the model is asked to predict bounding box

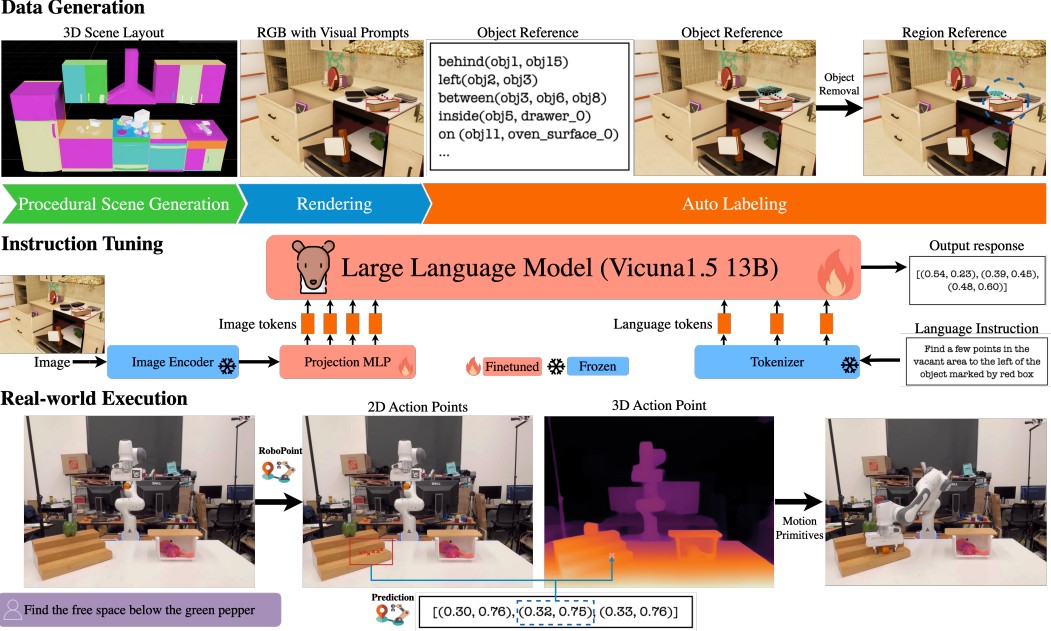

Figure 2: **Overview of ROBOPOINT pipeline.** An RGB image is rendered from a procedurally generated 3D scene. We compute spatial relations from the camera's perspective and generate affordances by sampling points within object masks and object-surface intersections. These instruction-point pairs fine-tune the language model. During deployment, ROBOPOINT predicts 2D action points from an image and instruction, which are projected into 3D using a depth map. The robot then navigates to these 3D targets with a motion planner.

center and dimensions for all instances of a certain category. This teaches the model how to ground language to image regions. The last two data sources, object reference and free space reference, are from our synthetic data pipeline (Sec. 4), where the object is to identify points on an object or a vacant region, satisfying certain spatial relations. These data enable the VLM to generate precise action points. We formulate different data sources into the same format and co-train with all of them. Table 4 evaluates the importance of each component in our data mix.

## 4 Dataset

We generate a diverse dataset in simulation by procedurally randomizing scene layouts, objects, and camera viewpoints. A novel aspect of our pipeline is generating affordance in free space, allowing the model to detect regions without distinct visual cues.

**Procedural Scene Generation in Simulation** To train ROBOPOINT, we generate a large photorealistic dataset in simulation annotated with affordance points. Most existing robotics datasets [47, 48, 49, 50] only have a handful of fixed artist-designed scene layouts which limits the types of relations that can be generated. We create a diverse dataset by procedurally randomizing several aspects of the scene: the 3D layouts, objects and camera view points. The scene is represented as a articulated body, including revolute (e.g. fridge, dishwasher doors) as well as prismatic joints (e.g. cabinet drawers). Objects are sampled from a large repository [51] with over 8K instances and 262 categories. The objects can be placed on any support surface. This allows our model to learn relations in a truly 3D environment. Once the 3D scene is created, we compute spatial relations among the objects and render an image for each relation from a diverse set of viewpoints in parallel. The relations include left, right, in front, behind, above, below, next to, and in between objects; on, on the left, right, front, and back parts of a surface; and inside a container. The diverse view distribution allow ROBOPOINT to maintain a consistent prediction across different viewpoints (Fig. 4). Around 660K (image, relation) pairs are generated from 10K scenes. Some examples from the dataset are shown in Table 1. More details can be found in Sec. B in the supplementary.

| Source | Object Reference | Free Space Reference | VQA [45] | LVIS [46] |
|--------|------------------|----------------------|----------|-----------|
| |  |  |  |  |
| Quantity | 347K | 320K | 665K | 100K |
| Query | Locate several points on an item situated beneath the bordered item. | Find some spots within the vacant space situated between the marked items. | What is the person feeding the cat? | Find all instances of cushions. |
| Answer | [(0.56, 0.69), (0.53, 0.76), (0.45, 0.72), ...] | [(0.57, 0.48), (0.58, 0.49), (0.56, 0.45), ...] | The person is feeding an apple to the cat. | [(0.49, 0.38, 0.08, 0.06), (0.53, 0.42, 0.07, 0.05), ...] |

Table 1: **Our dataset for instruction-tuning** combines object and space reference data with VQA and object detection data. ROBOPOINT leverages spatial reasoning, object detection, and affordance prediction from these diverse sources, enabling it to generalize combinatorially. Some answers are not shown in full due to space.

**Generating Affordance in Free Space**  A key novelty in our data pipeline is the generation of affordance in free space. This allows ROBOPOINT to detect regions without distinct visual cues, e.g. "the left part of pizza box" in Fig. 5, which an off-the-shelf object detector will not be able to detect. We employ a simple yet effective strategy. Namely, we first compute relations between a target object and another object or surface. Then, we remove the target object, re-render the image, and sample points inside the intersection of the target object mesh and the surface supporting it. This creates affordance labels in free space in relation to other entities in the scene.

## 5  Experimental Results

We demonstrate that ROBOPOINT achieves superior accuracy in spatial affordance prediction and real-world language-conditioned manipulation than state-of-the-art VLMs [21, 11] and visual prompting methods [17, 22]. Its view-point consistent prediction and conversational ability also enables application to navigation and augmented reality.

### 5.1  Spatial Affordance Prediction

ROBOPOINT significantly outperforms baselines in terms of accuracy on pointing to objects and free space referred by language. In addition, it generalizes to novel relation types, respects physical constraints, maintains common sense knowledge and produces view-consistent predictions.

**Benchmarks**  We evaluate spatial affordance prediction on two problems: object reference and free space reference. The object reference data is a 750-image subset of RoboRefIt [52]. Unlike human-centered dataset such as RefCoco [23], RoboRefIt features cluttered images with similar-looking objects that can only be distinguished by relational references.

Unlike object reference, no existing dataset addresses identifying *free space*. Therefore, we collect WHERE2PLACE, a dataset of 100 real-world images from homes and offices in the wild. To minimize bias, we ask one group to label each image with an instruction describing a vacant region relative to other entities, and a different group to label masks according to the instruction. As shown in Fig. 3, WHERE2PLACE features diverse and challenging scenes with clutter. A subset of 30 examples (WHERE2PLACE (h)) contain relation types not in our synthetic data. More details can be found in Sec. D in the supplementary.

**Baselines**  We compare ROBOPOINT against 3 state-of-the-art VLMs, Qwen-VL [6], LLaVA-NeXT [21], GPT-4o [11] as well as SpaceLLaVa [53], a community implementation of SpatialVLM [22]. We employ a zero-shot visual prompting strategy effective for pretrained VLMs. We label the input image with axes indicating its dimensions and ask the model to output a bounding box (top-left and bottom-right corners) of the target object/region, then sample evenly within the bounding box. For GPT-4o, we also tested in-context learning (GPT-4o-ICL) by providing 14 input-output pairs from our synthetic dataset as context before the query. In-context learning achieved zero accuracy for Qwen-VL and LLaVA-Next, likely because point outputs were not part of their training data.

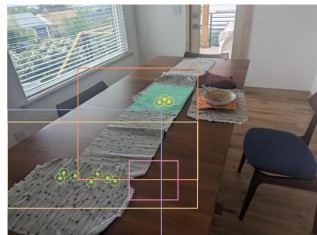 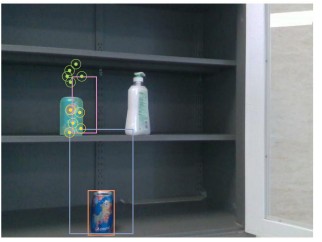 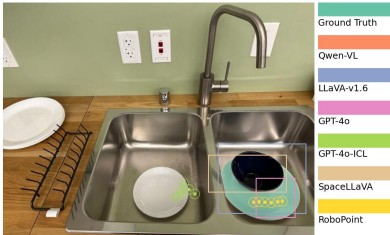

(a) left of bowl and on the tarp    (b) pepsi can on the middle shelf    (c) on the rightmost white plate

Figure 3: **Visualization of spatial affordance prediction on objects and free space.** ROBOPOINT can generalize to (a) combinations of seen relations; (b) unseen relations and (c) scenarios with physical constraints.

| | Qwen-VL [6] | LLaVA-NeXT-34B [21] | SpaceLLaVA [53] | GPT-4o [11] | GPT-4o-ICL [11] | ROBOPOINT |
|---|---|---|---|---|---|---|
| RoboRefIt [52] | $24.08 \pm 0.85$ | $19.91 \pm 0.92$ | $21.30 \pm 0.87$ | $15.28 \pm 1.27$ | $9.01 \pm 6.45$ | $\mathbf{49.82 \pm 0.52}$ |
| WHERE2PLACE | $10.49 \pm 0.77$ | $15.02 \pm 0.88$ | $11.84 \pm 0.73$ | $29.06 \pm 1.33$ | $14.46 \pm 6.38$ | $\mathbf{46.77 \pm 0.45}$ |
| WHERE2PLACE (h) | $9.90 \pm 0.22$ | $14.76 \pm 2.42$ | $12.10 \pm 1.36$ | $27.14 \pm 1.47$ | $14.83 \pm 4.68$ | $\mathbf{44.48 \pm 1.35}$ |

Table 2: **Quantitative comparisons on object reference (RoboRefIt) and free space reference (WHERE2PLACE).** ROBOPOINT outperforms state-of-the-art VLMs by a significant margin, even on examples where the spatial relations are unseen during fine-tuning (WHERE2PLACE (h)). The metric is percentage of predicted points within the target mask.

| | GQA [25] | MME [54] | POPE [55] | RefCoco [23] | SEED [56] | TextVQA [57] | VizWiz [58] | VQA-v2 [59] |
|---|---|---|---|---|---|---|---|---|
| LLaVA-13B [45] | 63.24 | 1522.59 | 85.92 | 31.99 | 67.06 | **48.73** | 56.65 | **78.26** |
| ROBOPOINT | **63.28** | **1524.78** | **86.01** | **32.16** | **67.52** | 47.31 | **60.37** | 77.83 |

Table 3: **Quantitative evaluation on standard VQA benchmarks.** ROBOPOINT performs on par with state-of-the-art VLM, maintaining the common sense knowledge learned from pretraining.

| No VQA [45] | No LVIS [46] | No Object Ref | No Space Ref | 10% Data | All |
|---|---|---|---|---|---|
| $28.28 \pm 2.08$ | $34.27 \pm 0.62$ | $42.23 \pm 2.28$ | $13.21 \pm 1.04$ | $15.71 \pm 0.77$ | $\mathbf{46.77 \pm 0.45}$ |

Table 4: **Ablation on the data composition.** Results on WHERE2PLACE show that best results are achieved when all of the data sources are combined during instruction-tuning.

**Results** In Table 2, we report the average prediction accuracy for ROBOPOINT and the baselines along with standard deviation computed from 3 different runs. The accuracy is calculated as the percentage of predicted points within the ground truth target mask. We can see that ROBOPOINT achieves significantly higher accuracy than all baselines, demonstrating the power of ROBOPOINT in spatial reasoning and precise target generation. Some results are visualized in Fig. 3.

**Does ROBOPOINT generalize to unseen relation types?** The synthetic dataset we constructed in Sec. 4 contains templated language and a fixed set of relations. Nevertheless, ROBOPOINT is able to produce accurate predictions for combinations of seen relations (Fig. 3a) and novel relation types such as in the middle, rightmost etc. that are not in the fine-tuning dataset (Fig. 3b). It is also able to maintain its advantage over baselines on these novel relations (Table 2).

**Does ROBOPOINT respect physical constraints?** ROBOPOINT's outputs not only satisfy the spatial relations but also respect physical constraints. The target points generated by ROBOPOINT avoid obstacles such as the the bowl in Fig. 3c, whereas the baselines fail to do so.

**Does ROBOPOINT keep common sense knowledge?** We evaluate ROBOPOINT's performance on VQA benchmarks and summarize the results in Table 3. ROBOPOINT performs on-par with LLaVA-v1.5-13B [45], a VLM trained on the same pre-trained weights as ROBOPOINT on VQA data. This shown that ROBOPOINT serves a generic VLM rather than a domain-specific model.

**How important is each component in the data mix?** In Table 4, we evaluated the importance of each data component on the WHERE2PLACE benchmark. Each data component – VQA on real images, object detection from LVIS, object and free space reference on synthetic images – significantly

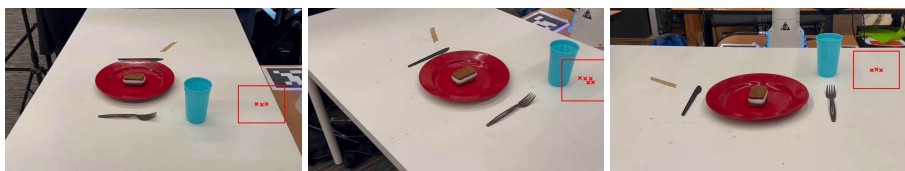

Figure 4: **ROBOPOINT's prediction is consistent across different viewpoints.** Red cross shows ROBOPOINT's response to "find free space right of the blue cup" in different views.

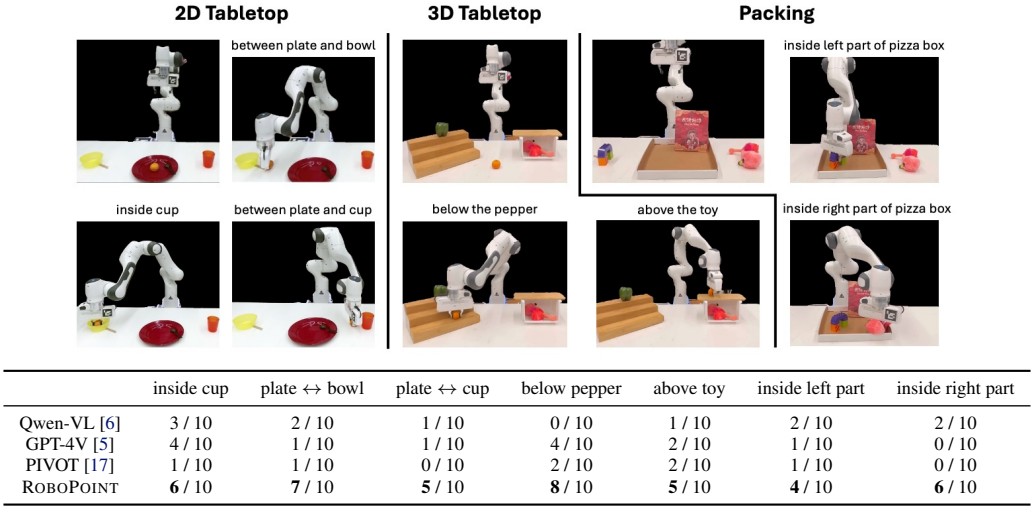

| | inside cup | plate ↔ bowl | plate ↔ cup | below pepper | above toy | inside left part | inside right part |
|---|---|---|---|---|---|---|---|
| Qwen-VL [6] | 3 / 10 | 2 / 10 | 1 / 10 | 0 / 10 | 1 / 10 | 2 / 10 | 2 / 10 |
| GPT-4V [5] | 4 / 10 | 1 / 10 | 1 / 10 | 4 / 10 | 2 / 10 | 1 / 10 | 0 / 10 |
| PIVOT [17] | 1 / 10 | 1 / 10 | 0 / 10 | 2 / 10 | 2 / 10 | 1 / 10 | 0 / 10 |
| ROBOPOINT | **6** / 10 | **7** / 10 | **5** / 10 | **8** / 10 | **5** / 10 | **4** / 10 | **6** / 10 |

Figure 5: **Real-world manipulation evaluation.** We created 7 language-conditioned manipulation tasks to measure ROBOPOINT's capability on real robot. ROBOPOINT outperforms the best baseline by 39.5% on average success rate, which depends critically on the alignment between the point predictions and the language.

contributes to overall accuracy. This highlights the value of a general problem formulation that incorporates diverse data sources. Additionally, data quantity is crucial, as the model's performance drops significantly when fine-tuned on only 10% of the data.

**Are RoboPoint's predictions consistent across views?** As shown in Fig. 4, ROBOPOINT maintains consistent predictions with camera movement. This makes it particularly suitable for mobile platforms and AR, where ROBOPOINT provides consistent action suggestions with moving cameras. Videos can be found on the project page robo-point.github.io.

### 5.2 Downstream Applications

To assess ROBOPOINT's capabilities on downstream robotics and vision tasks, we curated various scenarios for manipulation, navigation and AR assistance. We demonstrate ROBOPOINT's superior performance against state-of-the-art baselines on these tasks. Recordings of robot executions can be found on the project page robo-point.github.io.

**Real-World Manipulation** We set up 3 manipulation environments with 7 tasks (Fig. 5). The robot processes image observations and language commands through ROBOPOINT, which returns 2D point targets. These targets are converted to 3D points using a depth map (Fig. 2). The robot's end-effector pose is computed from these 3D points plus an offset. A motion planner then executes the trajectory to the target pose. Success is determined by collision-free execution and accurate placement of the target object as per the language instruction. We conducted 10 trials per task and compared ROBOPOINT against zero-shot VLMs like Qwen-VL [6] and GPT-4V [5], as well as iterative prompting methods such as PIVOT [17]. ROBOPOINT surpasses GPT-4V, the best-performing baseline, by a margin of 39.5% on average success rate. It also enables new capabilities. For instance, in the packing scene, ROBOPOINT's relational reasoning allowed the robot to differentiate regions within a pizza box, fitting multiple objects accurately.

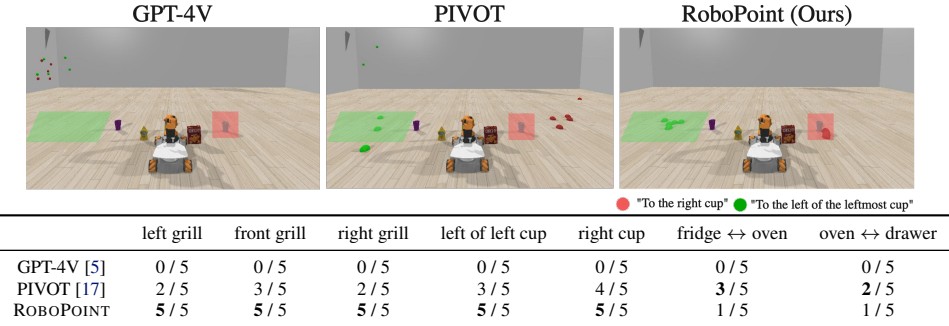

GPT-4V     PIVOT     RoboPoint (Ours)

● "To the right cup"  ● "To the left of the leftmost cup"

|  | left grill | front grill | right grill | left of left cup | right cup | fridge ↔ oven | oven ↔ drawer |
|---|---|---|---|---|---|---|---|
| GPT-4V [5] | 0 / 5 | 0 / 5 | 0 / 5 | 0 / 5 | 0 / 5 | 0 / 5 | 0 / 5 |
| PIVOT [17] | 2 / 5 | 3 / 5 | 2 / 5 | 3 / 5 | 4 / 5 | **3** / 5 | **2** / 5 |
| ROBOPOINT | **5** / 5 | **5** / 5 | **5** / 5 | **5** / 5 | **5** / 5 | 1 / 5 | 1 / 5 |

Figure 6: **Application to navigation.** ROBOPOINT predicts accurate goal point based on language, leading to higher target reaching rate than GPT-4V and PIVOT. Ground truths are drawn as colored masks and predictions are drawn as colored spheres.

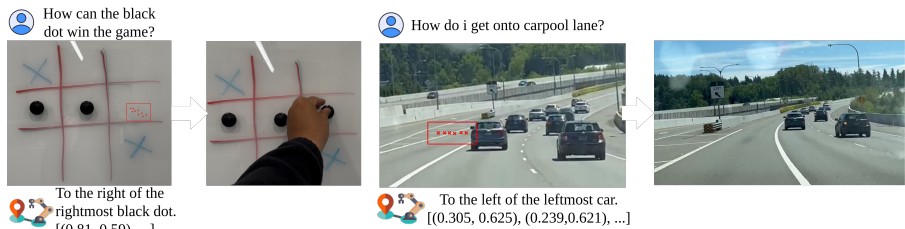

How can the black dot win the game?

To the right of the rightmost black dot. [(0.81, 0.59), ...]

How do i get onto carpool lane?

To the left of the leftmost car. [(0.305, 0.625), (0.239,0.621), ...]

Figure 7: **Application to Augmented Reality.** Given a user query, ROBOPOINT first generates natural language response using common sense and then provide visual guidance using spatial affordance prediction, which the user can execute with greater ease than language guidance.

**Navigation**   To evaluate ROBOPOINT's spatial affordance predictions beyond tabletop scenarios, we created 3 room scenes using the YouBot mobile manipulation platform in CoppeliaSim [60], where the robot is tasked to navigate to a target region with respect to certain entities in the scene. Fig. 6 shows the distribution of affordance generated by ROBOPOINT, PIVOT [17] and GPT-4V [5] and the success rate of navigating to the correct region using the predicted points with a simple path planner. ROBOPOINT outperforms PIVOT and GPT-4V in 2 out of 3 scenarios, demonstrating its effectiveness in large-scale room environments for navigation.

**Augmented Reality**   ROBOPOINT, which is co-trained with VQA data, retains conversational capabilities in natural language. As demonstrated in Fig. 1, users can interact with ROBOPOINT through language and receive action suggestions visually with the predicted affordance. In addition to the *set a formal dining table* task in Fig. 1. We demonstrate two more real-world scenarios-*win tic-tac-toe* and *get to carpool lane*-in Fig. 7, where ROBOPOINT gives visual guidance to solve the tasks by predicting the correct spatial affordance points.

## 6  Conclusion

We propose ROBOPOINT, a novel VLM designed to predict spatial affordances in images based on relational language instructions. By integrating real-world VQA data with automatically generated synthetic data, ROBOPOINT is able to generate precise action points that adhere to spatial and physical constraints, overcoming the limitations of current VLMs in robotics, which often rely on pre-defined motion primitives or large-scale expert demonstrations. Experimental results show ROBOPOINT's superior performance in complex tasks, such as relational free space reference and object rearrangement in cluttered environments, compared to state-of-the-art methods. Additionally, ROBOPOINT's versatility extends its applicability to augmented reality and robot navigation, showcasing its potential for broader applications in robotics.

**Limitation:**   ROBOPOINT does not provide confidence estimates for the point predictions. The number of output points are also not controllable. The points also do not provide a direction for action, which can be useful in manipulating articulated objects [61, 62]. All of these are valuable directions to explore in future work.

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

# Supplementary Material for ROBOPOINT

## A   Instruction Tuning

ROBOPOINT is instruction-tuned from a Vicuna-v1.5-13B base model [43] with a ViT-L/14 336px image encoder pretrained with CLIP [63]. The projector is a 2-layer MLP pretrained on the 558K subset of the LAION-CC-SBU dataset with BLIP captions from [7]. The instruction tuning took 40 hours on 8 A-100 GPUs with a batch size of 16 per-GPU. The learning rate is set to 4e-5.

## B   Data Generation

Table A shows more examples from our procedually generated synthetic dataset for object reference and free space reference.

We sample assets that one can find in an typical kitchen environments (e.g. dishwasher, hood, table, fridge) and use heuristics to place them in random, but semantic layouts in the scene. Once the furniture assets are added to the scene. We used a large object dataset sampled from ACRONYM [51]. Object positions are randomly sampled on support surfaces (e.g. countertop, table) and the orientations are determined by their stable poses. Poses that result in the object being in collision with the existing scene are rejected. We place cameras randomly in the scene and select those with at least three visible objects (visible means the number of points within segmentation mask is larger than 100) and at least 1 valid relationship between a pair of visible objects. The diverse view distribution allow ROBOPOINT to maintain a consistent prediction across different viewpoints. Around 660K (image, relation) pairs are generated from 10K scenes.

We use the 3D bounding boxes of objects, surfaces and containers in the scene layout to compute a set of pairwise relations, including left, right, in front, behind, above, below, next to, in between, on, on left part, on right part, on front part, on back part, and inside. Note that although these relations are templated, the model fine-tuned on these data is able to generalize to new types of relations. For each relation, there is a target object plus 1 or 2 reference objects. We generate 2 data instances for each relation, one for object reference and one for free space reference. The ground truth for object reference is a set of points sampled within the mask of the target object. Then, we remove the target object and re-render the image. The ground truth for free space reference consists of points sampled within the mask of the supporting surface of the target object in the re-rendered image with the target object removed. Sometimes the supporting surface is not visible due to occlusion and we filter out these instances. We use furthest point sampling starting from a random point in the mask, resulting in 1 to 50 ground truth points sampled per instance. We convert the sampled points to a list of image coordinates normalized between 0 and 1 and use that as the ground truth response.

One caveat for these procedurally generated scenes is that the objects do not have rich text descriptions. Most objects just have a category name. We get around this problem by adding visual prompts to the rendered images. Specifically, we draw colored bounding boxes around the objects referenced in the language instruction. As a result, a typical instruction in the synthetic data will look like: "There is an object surrounded by a red rectangle in the image. Find some places in the free area to the left of the marked object." Note that we do not add these visual prompts during testing, and thus do not require object detection. The idea is that the model learns to detect objects from other sources of data (e.g. LVIS [46]), and it will focus on relational reasoning when dealing with the object and space reference data.

## C   Qualitative Examples and Failure Mode Analysis

Fig. A provides additional qualitative comparisons of ROBOPOINT against baselines on RoboRe-fIt [52] and WHERE2PLACE. It also highlights two common failure modes. First, ROBOPOINT may fail to detect the correct reference object amidst challenging distractors. As seen in Fig. Ag, ROBOPOINT mistakes the black case for the black lid and incorrectly outputs points between the airpods and the black case. Second, ROBOPOINT may produce points that meet spatial relations but lack common sense, such as in Fig. Ah, where it places points "underneath the monitors" on the floor instead of on the table where a human would logically place them.

| Relation | Above | Behind |
|---|---|---|
| |  |  |
| Prompt | The image features an item encased in a red rectangular border. Locate several spots within the vacant space situated above the bordered item. | In the image, an object is framed by a red rectangle. Locate a few points on an object that is situated behind the framed object. |
| Relation | Between | Inside |
| |  |  |
| Prompt | In the image, there is an item framed by a red rectangle and another item encased within a green rectangle. Locate several points upon the item situated between the two highlighted items. | The image depicts a container delineated by a red rectangular border. Pinpoint several spots within the vacant area enclosed by the outlined container. |
| Relation | Right | On left part |
| |  |  |
| Prompt | The image features an object outlined by a red rectangle. Locate several points on an item that is situated on the right side of the marked item. | The image showcases an area demarcated by a red rectangle. Locate a few points within a vacant area on the left side of the marked surface. |

Table A: Examples from the synthetic dataset used to teach ROBOPOINT relational object reference and free space reference. The red and ground boxes are visual prompts to indicate reference objects and the cyan dots are the visualized ground truth (not included in the image inputs to the model).

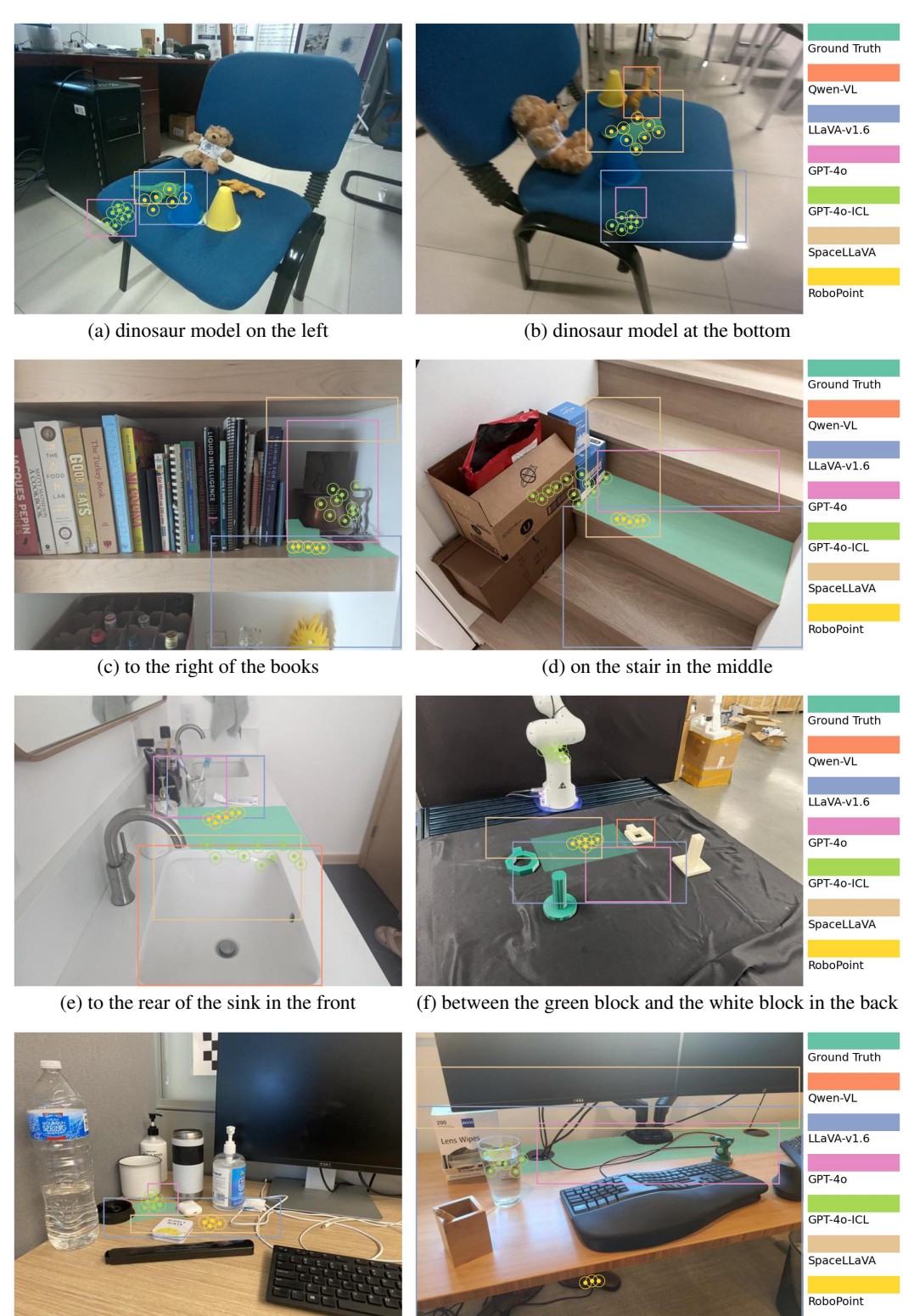

Figure A: Qualitative results on RoboRefIt (a, b) and WHERE2PLACE (c, d, e, f, g, h), including cases with relations unseen during training (d, e, f, h) and where GPT-4o performs better (g, h).

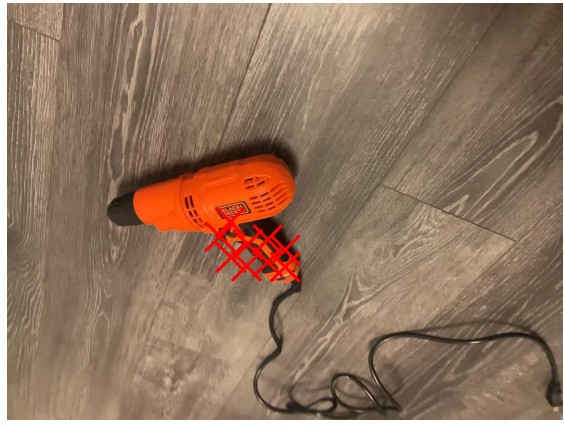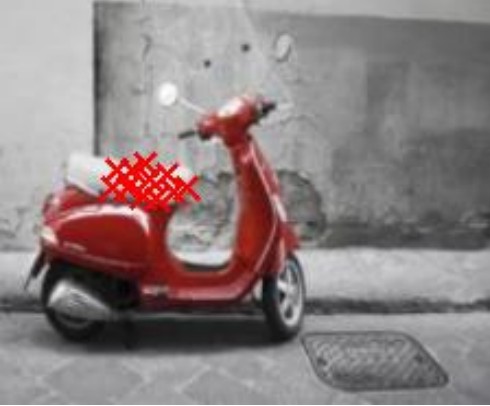

(a) Find several points on the handle of the power drill.        (b) Locate a few places to sit on the motorcycle.

Figure B: Some examples on part-based affordance prediction from (a) ManipVQA [64] and (b) AffordanceLLM [65].

## D WHERE2PLACE Benchmark

The process of collecting and annotating WHERE2PLACE consists of three steps, each conducted by different groups of people:

1. **Image Collection**: Images of random scene with multiple objects are taken. Each environment typically has only one image taken, with a maximum of five images captured per environment.

2. **Language Description**: Another group uses language to describe interesting object-region relationships. For example, "Find some free space in between the mouse and the keyboard."

3. **Mask Annotation**: The final group annotates the ground truth mask of the region corresponding to the language description using CVAT, an online annotation tool.

For both stage 2 (language description) and stage 3 (mask annotation), two people independently annotate each image and verify each other's annotations to ensure quality and consistency.

The scenes encompass a wide variety of environments, including homes, offices, and labs. Images are taken in different areas such as dining rooms, bathrooms, staircases, closets, with viewpoints ranging from top-down, sideways, close-up, to pan-out. The instructions include 3D relations like "on top of" and "below," as well as novel types of relations not present in our fine-tuning data, such as "leftmost" and "in the middle". More examples of WHERE2PLACE can be found in Fig. A.

## E Part Affordance Prediction

ROBOPOINT can also be applied zero-shot to locate object parts based on their function (affordance). Fig. B shows some examples. (a) is from ManipVQA [64] and (b) is from AffordanceLLM [65].

**Acknowledgments**

This project was partially supported by Amazon Science. We thank Yi Ru Wang, Tucker Hermans, Ajay Mandlekar, Jonathan Tremblay, Wei Yang, and Jie Xu for providing images and annotations in the WHERE2PLACE benchmark.

