# OpenReview forum: "RoboPoint: A Vision-Language Model for Spatial Affordance Prediction in Robotics"
_robot-learning.org/CoRL/2024/Conference — CoRL 2024_

### Official Review · Reviewer_zjRR · 2024-07-15
**Reviews on the RoboPoint**

**Originality:** 4
**Technical Quality:** 4
**Clarity Of Presentation:** 4
**Potential Impact:** 3
**Recommendation:** 4
**Confidence:** 5

**Review:**

## Strengths:

- The training data is mostly from the available open-sourced dataset and synthetic dataset, which greatly reduce the data collection cost.

- The author proposed to use point as a unified representation format to bridge the image space and action space. And this representation is generalized across different robotic tasks. Interesting designs.

- The method has been validated in different tasks.

## Weakness:

- The output is still limited in 2D image space, and the usage in the 3D world is still limited (still need the depth camera).

- Most of the demoed manipulation tasks are more on the pick-place ones, can this method be applied in the harder scenarios, e.g. articulated ones, as done in A3VLM[1] and GeneralFlow[2].

- The author mentioned the affordance, although the main focus is on the spatial affordance, but some experiments on the fine-grained semantic affordance as done in AffordanceLLM and ManipVQA, even the zero-shot one would be very interesting. Thought with the aid of LLM, such zero-shot ability could probably be also satisfying.


[1] Huang, Siyuan, et al. "A3VLM: Actionable Articulation-Aware Vision Language Model." arXiv preprint arXiv:2406.07549 (2024).
[2] Yuan, Chengbo, et al. "General flow as foundation affordance for scalable robot learning." arXiv preprint arXiv:2401.11439 (2024).

**Quality Of The Limitations Section:**

3

**Questions For Rebuttal:**

- See the Review Section.
- The author mentioned the number of points is not controlled, but how is the statistics?
- And can you provide the distribution of key points, would high variance exist?

**Robotics Focus:**

4

**Summary Of Paper:**

The paper proposes RoboPoint, a VLM that predicts image keypoint affordances given language instructions.

**Summary Of Recommendation:**

Very good paper overall, with some modifications needed to improve.

---

### Official Review · Reviewer_px3Y · 2024-07-16
**Review of Paper 138**

**Originality:** 3
**Technical Quality:** 3
**Clarity Of Presentation:** 2
**Potential Impact:** 2
**Recommendation:** 3
**Confidence:** 3

**Review:**

*Strengths*:
 - Within the now relatively cluttered space of VLMs for robot perception, this work finds a nice niche. Predicting keypoints in images seems like a useful objective.
 - The model seems very adept at reasonable about spatial relationships from single images.

*Weaknesses*:
 - This field is growing rapidly, yet there seem to be some recent references that seem very relevant but missing. [Spatial VLM](https://spatial-vlm.github.io/) in particular seems relevant but there are also a host of other methods that combine VLMs with semantic segmentation that it would seem could do something quite similar to what is proposed here with slight modifications.

 - One of the crucial aspects of the paper is the creation of the new dataset. Yet I have trouble to understand exactly how this was done in general. Some examples are given, such as the object removal heuristic, but is this trick always used? What are the set of spatial relations that are included in the dataset (perhaps this is in an appendix but I feel it deserves to be in the main manuscript)

- If I understand correctly, only the RoboPoint method is fine-tuned on the Object Reference and Free Space Reference datasets. Would it not be possible to fine-tune some of the baseline methods using these datasets aswell?


*Other more minor questions/comments*:

 - Why does the model output several points and how is the number of points chosen? In the figures, how do we interpret that red box, was that added manually?

 - I find the repeated use of the term "affordance" somewhat questionable. Certainly the model demonstrates the ability to perform some level of spatial reasoning, but the bar for "affordance" is much higher and I don't believe that it is really met here.



*Minor*:

 - L23: no comma needed here: "psychology, demonstrate". There are other places where commas could be removed.
 - You should define what you mean by "instruction-tuned" before using it.
 - Something wrong here: "pretrained primitives skills or external detection modules" (either need commas or not plural)
 - is the acronym VQA ever defined?
 - I'm confused by this statement: "Compared to ap- proaches that generate entire trajectories [14, 15, 16], data for spatial affordance prediction can be generated independently for each time-step" - do mean indepenence in the statistical sense? If so, this seems highly questionable.
 - Something wrong here: "points to target regions conditioned language commands" or hard to parse
 - L69: " a object" -> "an object"
 - Several places: "requires" -> "require"
 - Fig. 6 caption error: "0.0.81"


-----------
Post-rebuttal.

The authors have done an excellent job at addressing my concerns and I believe that the updated manuscript can be presented at CoRL. I have updated my rating accordingly.

**Quality Of The Limitations Section:**

2

**Questions For Rebuttal:**

I have 2 main points of clarification for the rebuttal (details above):

1. I would like more details about how the dataset was generated
2. I would like to know exactly which data was used to train / fine-tune each of the baselines
3. I would like further justification that this is the right set of baselines to compare against.

**Robotics Focus:**

3

**Summary Of Paper:**

The paper proposes RoboPoint - A VLM-basd model that takes language queries that include spatial reasoning and an image, and outputs keypoints that respond to the query.

**Summary Of Recommendation:**

I find the work interesting, however there are some important open questions about the dataset and experimental evaluation which I feel need to be addressed.

---

### Official Review · Reviewer_Wr3v · 2024-07-21
**Good novel and versatile approach for robotic control**

**Originality:** 3
**Technical Quality:** 4
**Clarity Of Presentation:** 5
**Potential Impact:** 3
**Recommendation:** 3
**Confidence:** 3

**Review:**

Quality: I think the paper is high-quality research with a well-designed methodology, comprehensive experiments, and great analysis. The authors have clearly put significant effort into developing a novel approach and evaluating it rigorously.

Clarity: The paper is generally well-structured and clearly written. The figures and tables effectively support the text.

Originality: Robopoint presents a novel approach to spatial affordance prediction for robotics, combining vision-language models with a point-based action space. The demonstrated effectiveness of the synthetic data generation pipeline is also innovative.

Significance: This work demonstrates significant potential impact, potentially bridging the gap between high-level language instructions and low-level robotic control in a more flexible and generalizable way than previous approaches.

Strengths:
- Novel approach: The point-based action space is a good solution to the precision problem in language-guided robotics.
- Scalable data generation: The procedural synthetic data generation pipeline allows for diverse and large-scale training data.
- Versatility: Robopoint demonstrates effectiveness across various robotic applications without domain-specific training.
- Strong performance: The model outperforms state-of-the-art baselines on multiple tasks.
- Generalization: Shows ability to handle unseen relation types and maintain common-sense knowledge.
- Practical implications: The approach has clear potential for real-world robotic applications.
Weaknesses:
- Limited exploration of failure cases: could benefit from a more in-depth analysis of scenarios where ROBOPOINT struggles or fails.
- Lack of confidence estimates: As acknowledged by the authors, the model doesn't provide confidence measures for its predictions, which could be crucial for real-world applications. Also the number of output points is not controllable, which could be a limitation in some scenarios.
- Relatively small real-world dataset: While the synthetic dataset is large, the real-world WHERE2PLACE dataset is relatively small (100 images).

Overall, despite these weaknesses, the paper presents a good contribution to the field with a novel, well-executed approach that is promising for language-guided robotics.

POST-REBUTTAL

The authors addressed my concerns, and I recommend accepting the paper. Thank you for the information about the dataset, and overall I think that this is a creative approach that has good potential and versatility for robotic tasks.

**Quality Of The Limitations Section:**

3

**Questions For Rebuttal:**

Despite the fact that performance on the object and free space reference tasks is less than 50%, there is very limited discussion on failure modes or arrow analysis with regards to the proposed approach. Could you provide specific examples of failure cases, and an analysis of common failure types?
What kinds of scenarios does the approach find challenging, and what are potential explanations for observed failure modes?
In regards to the proposed dataset, can you provide more comprehensive information about the WHERE2PLACE dataset?
What was the process for collecting and annotating this dataset?
How diverse are the scenes and instructions in the dataset?
How did you ensure the quality and consistency of the annotations?
Could you give more details about what the annotators were told to do?
Can you provide more details and examples of the types of instructions and scenes included in the dataset?
Also, what are the computational and time requirements for training or deploying the approach?
Could you also provide similar details about the time and resources used to generate the WHERE2PLACE benchmark?

**Robotics Focus:**

4

**Summary Of Paper:**

This paper introduces Robopoint, a novel VLM trained specifically for spatial affordance prediction in robotics. The key idea is to frame the problem as predicting 2D image coordinates such that they specify spatial relations described by a given language instruction.  Main contributions: New approach to robotic control; using point coordinates as output rather than high-level language plans or pre-determined primitives; Data generation pipeline; procedural method to generate large-scale synthetic data for training affordance models; Performance: Robopoint outperforms existing VLMs and other visual prompting techniques on spatial affordance prediction tasks, including the Where2Place benchmark introduced by the authors; Versatility: authors demonstrate that the approach is effective across a variety of robotic applications such as manipulation, navigation, AR, without domain-specific training

**Summary Of Recommendation:**

Robopoint presents a novel and promising approach to spatial affordance prediction in robotics. Its strengths lie in the innovative point-based action space, scalable synthetic data generation, versatility across applications, and strong performance compared to baselines. The work shows significant potential for advancing language-guided robotics. However, there are areas for improvement. The authors should address: More in-depth analysis of failure cases and challenging scenarios; Limited size of the real-world dataset (WHERE2PLACE). Despite these limitations, the overall quality, originality, and potential impact of the work are high. The authors should provide more details on the WHERE2PLACE dataset creation process, computational requirements, and expand on failure mode analysis. Addressing these points would further strengthen it.

---

### Author Rebuttal · Authors · 2024-08-13

We have attached a revised version of our submission and supplementary material incorporating the reviewers' feedback. The changes are highlighted in blue.

---

### Decision · Program_Chairs · 2024-09-04

**Decision:**

Accept

**Comment:**

This paper proposes to train a VLM that predicts image keypoint affordances given language instructions to solve robot manipulation problems.

The paper is well motivated, timely, tackles a relevant problem for the robotics community, contains sufficient novelty, is well written, and sufficient experiments are provided.

The authors addressed the main weakness of the paper, namely missing details, in the rebuttal phase. The authors should make sure to include these changes in the final version of the paper.

I recommend acceptance.